# Impact of Tourist's Environmental Awareness on Pro-Environmental Behavior with the Mediating Effect of Tourist's Environmental Concern and Moderating Effect of Tourist's Environmental Attachment

**Shahrukh Aman** [1], **Nadir Munir Hassan** [1], **Mohammad Nisar Khattak** [2,*], **Mohamed A. Moustafa** [3,4], **Mahendra Fakhri** [5] and **Zeeshan Ahmad** [1,*]

1.  Department of Business Administration, Multan Campus, Air University, Multan 60600, Pakistan; Shahrukhaman161@gmail.com (S.A.); nadir.magsi@aumc.edu.pk (N.M.H.)
2.  Department of Management, College of Business Administration, Ajman University, Ajman 346, United Arab Emirates
3.  Department of Management, College of Business Administration, King Faisal University, Al-Ahsa 31982, Saudi Arabia; mamoustafa@kfu.edu.sa or mohamed.ali.yousef@fth.helwan.edu.eg
4.  Tourism Studies Department, Faculty of Tourism and Hotels Management, Helwan University, Cairo 11795, Egypt
5.  Department of Business Administration, Faculty of Communication and Business, Telkom University, Bandung 40257, Indonesia; mahendrafakhri@telkomuniversity.ac.id
*   Correspondence: Muhammad.khan@ajman.ac.ae (M.N.K.); zeeshan.ahmad@aumc.edu.pk (Z.A.); Tel.: +971-54-410-9322 (M.N.K.); +92-337-790-6906 (Z.A.)

**Abstract:** Pakistan has been blessed with rich tourism potential because of its rich history, culture, biological and geographical diversity. Travelers have for quite some time been attracted to Pakistan to encounter a nation that flaunts rugged natural beauty, cultural richness, and unparalleled hospitality. Pakistan has various tourist destinations in the northern areas of Pakistan. Kumrat Valley has become a tourist hotspot in recent times after the prime minister of Pakistan, Mr. Imran Khan, visited the valley. To reduce the negative effects on these tourist destinations due to the accelerating increase in tourists, the present study used a quantitative approach to uncover whether there is an environmental awareness–behavior gap among the tourists, with their level of environmental awareness outweighing pro-environmental behavior. Using a sample of 426 tourists who have visited the valley, the analysis of the results indicates that the pro-environmental behavior is positively and significantly affected by the components, environmental awareness, environmental concern, and environmental attachment. Environmental attachment is added as a moderator between environmental awareness and pro-environmental behavior. This study suggests that environmental awareness must be promoted among tourists to ensure that they exhibit pro-environmental behavior.

**Keywords:** environmental awareness; environmental concern; pro-environmental behavior; environmental attachment; Pakistan tourism

## 1. Introduction

Over the last 50 years, the travel and tourism industry has become an important worldwide economic segment that has gone through huge development [1]. The worldwide economic role of the tourism industry has moreover kept on expanding with the World Travel and Tourism Council assessing the sectoral commitment to worldwide economy in 2018 at USD 7.2 trillion (9.8% of global GDP) and 284 million jobs (9.1% of jobs worldwide) [2]. The significance of the travel industry economy is much more articulated in a large number of objective networks and the in excess of 90 nations where the travel industry speaks to over 10% of public Gross Domestic Product (GDP) and a critical extent of the business [2]. Similar is the case for a country such as Pakistan. Pakistan has been blessed

with rich tourism potential because of its rich history, culture, biological and geographical diversity. Travelers have for quite some time been attracted to Pakistan to encounter a nation that flaunts rugged natural beauty, cultural richness, and unparalleled hospitality. Pakistan has various tourist destinations in the northern areas of Pakistan, such as at Swat, Malam Jabba, Balakot, Kaghan, Naran, Chitral, Gilgit-Baltistan, Hunza, etc. The northern areas are famous for its mountainous ranges, natural beauty, historical and archaeological sites. Tourists always like to travel towards popular destinations.

Kumrat Valley has become a tourist hotspot in recent times, when Mr. Imran Khan (Prime Minister of Pakistan) mentioned Kumrat Valley as one of the most beautiful places to visit in the northern areas. As this valley was underrated before, tourists started heading towards Kumrat Valley. Kumrat is covered with fields full of greenery, mountains covered with snow, the water-canal Panjkora; clouded hills and woodlands are the charms of the valley, which occupy living spaces for a combination of a variety of flora and fauna. It is in the Upper Dir Kohistan district at the behind of which Swat Kohistan zone of Gabral is found. An element of Kumrat Valley is its transcending Deodar woods trees situated on level ground neighboring the Panjkora stream. Subsequently, every late spring season a great many travel from various regions of the country to visit Kumrat Valley for its greenery and cool climate, making it one of the most well-known tourists spots. Though the numbers of tourists are increasing in Kumrat Valley till now, it is safe from pollution, and tourists visiting this destination are showing a good behavior. The other tourist destinations are facing challenges such as the tourists are littering their waste thus causing pollution in the surrounding area. Littering causes pollution and changes the climate.

The travel and tourism industry can play an important role in handling such climate/environmental changes [3]. The travel and tourism industry is very sensitive due to its high dependence on good and quality climate of the destination [4]. The environmental deterioration caused by tourism and the lack of specialized end-to-end conservation management contributes to a decline in many tourist destinations [5]. The major environmental issues include air/water pollution, natural resource depletion, climate change, waste disposal, biodiversity loss, and overcrowding [6]. It can be assumed that tourists prefer a healthy environment that is contamination free, then the tourists are tempted to act in an environmentally hostile way [7]. Since such environmental challenges and problems in the tourism can be attributed to the actions of the tourist, it is crucial to acknowledge that individual's behaviors would affect the climate. Thus, research about raising public consideration to the environmental sustainability problems and from human behavior the impact on environment has started to receive substantial literary focus [8].

Literature has suggested that the significance of tourists' pro-environmental behaviors (PEBs) in improving life quality, reducing environmental footprint, and bringing high place attachment, thus highlight the positive impact of pro-environmental behavior on the lives of people. Pro-environmental behavior refers to the individuals' purposeful actions that can lessen negative environmental impact [9]. Pro-environmental behavior is the effort of an individual to minimize the damaging effect due to the destruction of nature by enriching and conserving the environment. It describes the exploits of a person or a group of people who encourage the sustainable use of natural resources [10]. The travel industry partners should be concerned about the natural issues of vacationer locales and participate in favorable ecological practices to continue climate and the travel industry. Researchers have received different terms to portray practices that ensure the climate, for example, naturally concerned practices, ecological criticalness of practices, earth dependable practices, and supportive of natural practices [11]. Pro-environmental behaviors are viewed as the precaution activities of people to secure the general climate by understanding nature, which addresses the natural issues [12]. As per sympathy with a travel industry site, occupants secure the climate to deflect harm. Such practices incorporate intentionally lessening contamination created by day-by-day life.

Environmental change represents a danger to the worldwide economy [13]. The threats that environmental change postures to development in the tourism industry and

its capacity to add to the SDGs specifically stays under explored [14]. There is extremely restricted exploration on the travel industry and environmental change with regards to less developed nations [15]. In spite of developing sectoral consciousness of the weakness of the travel industry to environmental change [16], the distinguishing changes in the environmental effects which are being tackled by the travel and tourism industry at the territorial and the scale for destination country still remain uncertain.

The literature has shown that the main roots of environmental problems are the behavior of people [17]. However, if the users change their behavior and their ways of how to interact with the product, it would be beneficial for the environment [18]. Behavioral change that benefits the environment can be referred to as pro-environmental behavior [19], which is a form of individual action that is directly related to environmental improvement. Consequently, it seems reasonable to induce pro-environmental behavior in these actors to minimize negative environmental externalities [20].

Despite the fact that the pro-environmental behavior concept has started to receive rising scholarly and managerial attention [21], questions that are about the foremost aspects that can tempt the people to embrace pro-environmental behavior increasingly occupied researchers' interests. There is an environmental awareness–behavior gap, with the level of environmental awareness outweighing pro-environmental behavior [22].

Notable scholars [23] have also suggested the significance of environmental awareness (EA) in explaining individuals' pro-environmental behaviors. Environmental awareness that provides people with an insight into the environmental implications of their decisions is also recognized as the primary step to getting the people ready for environmental issues [24]. High environmental awareness lets tourists' comply in a sustainable way [25]. Equally, little knowledge may also result in obliviousness; the lack of such awareness may lead to indifference, minimal personal behavior modifications, and dependence on government actions. Therefore, environmental awareness is one of the main premises of the adoption of pro-environmental behavior. Despite the importance of tourists' involvement in pro-environmental behavior, it remains unclear how environmental awareness is related to tourists' pro-environmental behaviors.

The environmental consciousness, which represents the concern and awareness of people about the environmental effects of their actions, is widely recognized as the first important step in the planning for environmental solutions. [24].

The current study is a significant addition to the swelling body of knowledge on environmental concerns and pro-environmental behaviors exhibited by the tourists, both domestic and international. Therefore, the objective of this paper is to explore the unique and novel construct from the perspective of a developing country and to explore the intervening mechanism of a tourist's environmental attachment. Secondly, the research integrates the ABC model by Albert Ellis (1995) to examine the disparity in the attitude–behavior of tourists visiting the north of Pakistan, specifically Kumrat Valley.

## 2. Theoretical Background

Detailed theoretical review concluded that pro-environmental behavior and environmental awareness are key factors towards a healthy environment in the sector of tourism in Pakistan. This study examined the mediating role of environmental concern between environmental awareness and pro-environmental behavior. This study also examined the moderating effect of environmental attachment.

### 2.1. Environmental Awareness (EA)

The studied literature review divides it into two streams: the first talks about a set of affections and understanding related to human behavior and environment [26], whereas the second describes it as an ability to integrate sensations from the environment and related issues with targeted goals to generate pro-environmental behavior [20]. Environmental awareness refers to tourists environmental awareness and the desire to influence the environment positively and to consider environmental concerns and their causes [26].

Environmental awareness represents individual interest and is known as the first significant step in people's readiness to address environmental issues, the understanding of the environmental consequences of their behavior [24]. People with a high degree of environmental consciousness are expected to be environmentally sustainable [25]. Growing environmental awareness among tourists can reduce pollution in developing countries [27]. Environmental awareness is an awareness of environmental issues that has a significantly increased environmental impact on individual behavior [28].

### 2.2. Environmental Concern (EC)

Environmental concern discusses a person's attitude towards the general or global environment, towards its decline, and many other issues on green environment [29]. Similarly, in tourist behavior [30], described environmental concern as attitude of the tourist who is worried about the environmental threats. The attitudes of travelers and behaviors of travelers is commonly used for studying tourist environmental issues in the tourism literature. The literature has indicated that the importance of environmental concern was because of the environmental knowledge and information about environmental problems and the problems raised by activities not eco-friendly and highly appreciated by a person, which concern the environment and its safety [31].

### 2.3. Pro-Environmental Behaviors (PEBs)

Tourists' pro-environmental behavior is the behavior of individual tourists or group of tourists contributing towards sustainable natural resources consumption [32]. The author used various words to refer to actions with beneficial effects on the environment. These included environmental-conscious behavior, eco-activism, green consumption behavior, and environmental behavior. In addition to the growing interest in the climate, environmental conditions have flourished over the years. However, attempts to explain the concepts are not very frequent. If we take this problem into account, we first give a description of pro-environmental behavior and speak about the characteristics and essence of various pro-environmental behavior.

Pro-environmental behaviors refer to the actions of an individual that can reduce negative effects on the environment [33]. Pro-environmental behaviors include a variety of behaviors such as recycling [34], transport use [35], waste management [36], energy consumption [37]. The present research is interested in studying the travelers' pro-environmental behaviors, which refers to the behaviors promoting the protection of the environment and ecosystems harming avoidance and also selecting the modes or products which are environmentally friendly [11].

### 2.4. Environmental Attachment (EAT)

Environmental attachment primarily addresses an emotional problem when people appreciate the environment and an emotional character that causes people to understand the fundamental value of the environment and show an environmental sense of exploration, gratitude, compassion, and guilt [38]. Environmental attachment with the natural places provides advantages to the nature in the restoration of positive emotions and psychological restoration [39]. Highly involved tourists have the tendency to dive into the local climate, thus enabling environmental attachment formation [40]. The creation of meaningful spaces for people to connect in public areas in the city created an environmental attachment. Lasting engagement in various cultures has a direct impact on the environmental attachment in cultural tourism destinations [40].

### 2.5. Supportive Theories
ABC Model

The first three letters are A for activating event, B for beliefs about the events, and C for consequences; these are often referred to as the ABC model [41]. Among various models proposed by various schools of thought to measure attitude, the ABC model, developed by

Solomon, is the most commonly used model to measure attitude toward an object [42]. The ABC model can represent attitudes emerging or occurring simultaneously, and can explain the correlation between environmental situation and emotional reaction or behavior [43]. This model is believed as the antecedent of attitude as one's mental readiness to respond positively or negatively toward an object such as people, place, or event and it consists of three components [42]. The beauty of this model lies in predicting how individuals feel or will react towards a particular object/thing or event. In our study, environmental awareness is the activating event (A), as the awareness of an individual can help in reducing the pollution in his nearby environment. After creating awareness in the mind of an individual, the sense of environmental concern is developed then from which an individual forms a belief (B) to protect the environment and becomes environmentally concerned which results in the consequences (C) on the pro-environmental behaviors due to this belief. The consequences can be positive or negative according to the pro-environmental behavior.

## 3. Hypothesis Formulation

### 3.1. EA and PEBs

The paper has shown that travelers are more likely to exhibit environmentally friendly or green behavior when they recognize environmental and ecological issues. For example, people who buy products with eco-friendly labels are more environmentally conscious, eat organic fruits, and are interested in recycling activities [44]. These behaviors inhibit the influence of the tourist's hazardous acts on a tourist spot and its environmental problems [45]. Moreover, travelers' environmental attitudes have been found to strongly influence their intentions to involve in environmentally friendly tourism [46]. Environmental awareness motivates them to exhibit environmentally responsible behavior. Hence, environmental awareness has been suggested to positively influence tourist pro-environmental behaviors [25]. A tourist's pro-environmental behavior is influenced by level of environmental awareness. If the awareness of a tourist is high, then the level of pro-environmental behavior is also high. According to this, it can be hypothesized that:

**Hypothesis 1 (H1).** *EA positively influence Tourists PEBs*.

### 3.2. EA and EC

Environmental awareness requires not only environmental awareness, but also the mindset, beliefs, and skills of a person to solve environmental problems. In addition, environmental awareness is the first step to carry out responsive behavior [47]. People across the globe are becoming conscious of their surroundings [30]. Environmental concern involves tourists decision to make environmental friendly choices [48] and intention to perform behaviors that are friendly towards the environment [49]. When individuals have appropriate environmental awareness, they become more concerned with regard to environment protection and sustainability [44]. The positive relationship existed between awareness and development of environmental concern [50]. Hence, travelers' environmental awareness has been suggested to positively influence their environmental concern. Accordingly, it can be hypothesized that:

**Hypothesis 2 (H2).** *EA positively influence EC*.

### 3.3. EC and PEBs

Environmental concern, also known as an environmental belief or pro-environment attitude, refers to the awareness of an individual about the importance of conserving the environment. Environmental concern has been an important indicator of many visitor behaviors, and has, therefore, the opportunity to explain why and how environmental concern has to do with the widespread protection of cultural and natural resources [51]. Following this theme, the literature has suggested the significance of environmental attitudes in predicting environmentally responsible behaviors [52]. For instance, environmental

attitudes and motivations such as environmental concern significantly influence an individual's green behavior [53]. Research indicates that the intense concern of individuals with respect to environmental concerns and how to cope with these problems contributes to behaviors. Scholars have identified multiple impacts of environmental concern on numerous pro-environmental behaviors [30]. Moreover, notable scholars [54] have suggested environmental concern to be of significant value to develop an understanding of pro-environmental behaviors, citing that concern has a strong positive impact on behaviors. The individuals with higher environmental concern have higher tendency to exhibit pro-environmental behaviors [55]. As more studies focus on the inter-relationships between environmental concern and pro-environmental behaviors, researchers find that people define environmental problems based on their subjective evaluations [56]. Hence, environmental concern is regarded as a requisite of pro-environmental behaviors. Accordingly, it can be hypothesized that:

**Hypothesis 3 (H3).** *EC positively influence PEBs.*

### 3.4. EC as a Mediator

Despite the observational proof connecting environmental awareness to environmental concern and thus to pro-environmental behaviors, which has been examined above, writing has not proposed and researched the intervening part of environmental concern in the connection between environmental awareness and pro-environmental behaviors. Whereas earlier literature has indicated the vital function of environmental concern as a mediator in the connection between environmental awareness and different tourist's conduct and attitudinal results. For example, environmental concern has been found to be mediating the relationship between awareness and green purchasing [50]. Similarly, Sadiq (2019) reported the mediating role of environmental concern between tourist optimism and pro-environmental consumption behaviors. Another study [50] also suggested the mediating role of environmental concern in the relationship between environmental knowledge and purchase intention. Based on these research findings, current study proposes tourist's environmental concern to act as a mediator between environmental awareness and pro-environmental behaviors. The relationship between the sense of individual responsibility and concern about its outcomes (i.e., EA) and the sense of involvement of the group will enable their moral responsibility to achieve pro-environmental behavior [26]. Thus, environmental concern, which can be tourists concern regarding environmental issues, may work as a mediating link between environmental awareness and pro-environmental behavior. The following hypothesis is formulated based on these discussions:

**Hypothesis 4 (H4).** *The relationship between EA and PEBs is mediated by EC.*

### 3.5. EAT as a Moderator

Environmental attachment basically alludes to an enthusiastic characteristic of people whereby they like the indigenous habitat and a passionate quality that drives people to perceive the inborn estimation of the climate, mirroring a feeling of natural disclosure, thankfulness, sympathy, and blame [38]. These enthusiastic and psychological components of favorable to natural connection are identified with more crucial supportive of ecological qualities and—without relevant imperatives—will in general be prescient of pro-environmental behavior [57]. Environmental attachments play a major role in tourist pro-environmental behavior, as shown by [58]. Refs [59,60] found evidence that the emotional influences of tourism's pro-environmental behaviors were found to be vital to the environmental attitudes and sustainable tourism [61,62], including natural environment sentiments. Considering environmental attachment a solid indicator for pro-environmental behaviors in various settings, subsequently, we propose natural connection—a solid earlier passionate inspiration—might be essential for the overflow of pro-environmental behavior across settings, given the dedication needed to conquer persuasive and logical

hindrances [63]. In other words, individuals with higher environmental awareness may put forth more attempt to concede to act favorable to earth across settings. We, therefore, start with the following hypothesis:

**Hypothesis 5 (H5).** *The relationship between EA and PEBs is moderated by EAT.*

Figure 1 shows the theoretical framework based on the proposed hypothesis.

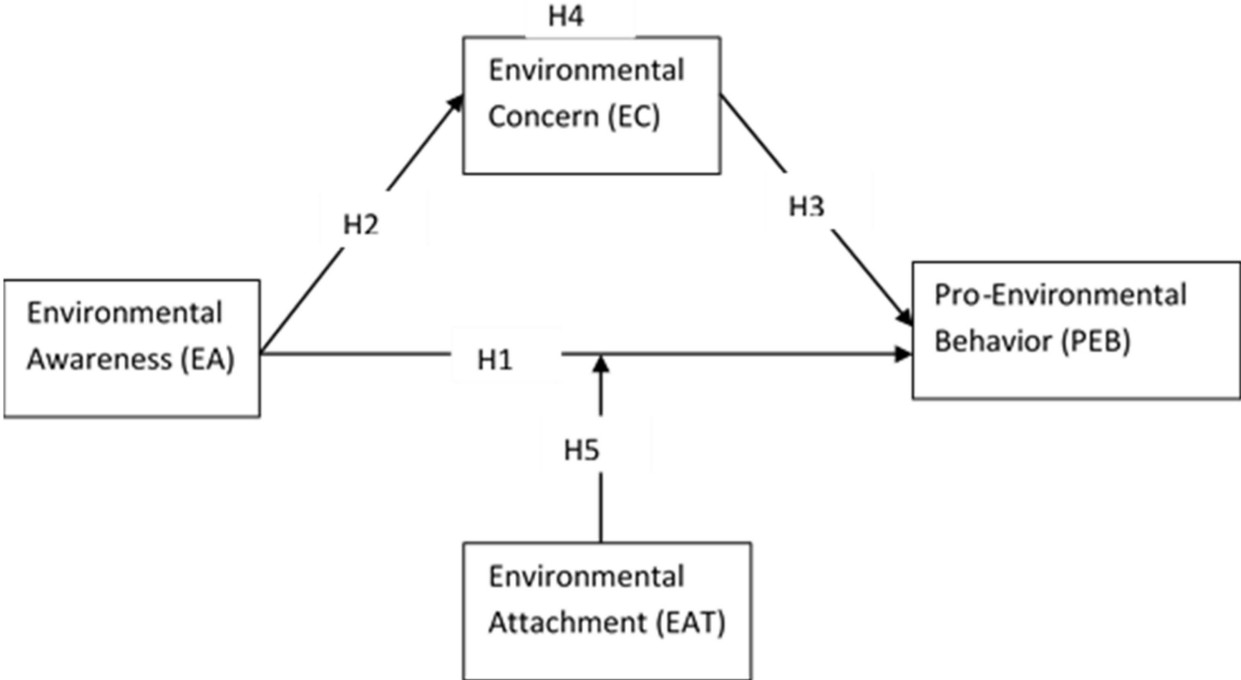

**Figure 1.** Theoretical Framework.

## 4. Materials and Method

### 4.1. Data Collection and Sample Characteristics

For the present research we selected Kumrat Valley as the tourists visiting this valley are showing a positive response towards maintaining its natural stature. It is located in the Upper Dir district of Khyber Pakhtunkhwa province (Pakistan). The valley of Kumrat is among the most beautiful valley, which is enriched with natural resources and biodiversity, attracting thousands of tourists every year. Above all, the travel industry has placed immense strain on its natural resource base, combined with the realistic exploitation by visitors, tour operators, and local communities of natural resources (WWF-Pakistan, 2019). Many tourists from the local country as well as all over the world are visiting the northern areas of Pakistan. One of the hotspot areas for the tourists in Pakistan is Kumrat. People from different cities of Pakistan are visiting Kumrat Valley as it is a new tourist sensation. The researchers personally visited the Kumrat Valley to collect data during July–August–September 2020. Before collecting data, researchers had informed the participants regarding the study purpose and data confidentiality. Next, it was plainly passed on to the respondents that their obscurity will be guaranteed. Respondent were additionally educated and guaranteed that there will be no set-in stone responses to things. Using the convenience sampling of about 500, questionnaires were distributed; out of which, 436 were filled and returned, and 10 of them were incomplete, due to which they were discarded. Table 1 shows the demographic stats of the respondents.

**Table 1.** Classification of the sample characteristics.

| Variable | Categories | Frequency | Percentage (%) |
|---|---|---|---|
| Gender | Male | 239 | 56.1 |
| | Female | 187 | 43.9 |
| Age | Below 18 | 47 | 11 |
| | 18–28 | 197 | 46.2 |
| | 28–38 | 130 | 30.5 |
| | 38–48 | 46 | 10.8 |
| | Above 48 | 6 | 1.4 |
| Education | Intermediate | 63 | 14.8 |
| | Bachelors | 123 | 28.9 |
| | Masters | 132 | 31 |
| | M.Phil | 84 | 19.7 |
| | PHD | 24 | 5.6 |

*4.2. Measures*

Five-point Likert scale has been employed to assess the study variables (EA, EC, EAT and PEBs). EA was measured by employing a 3 items scale developed by Bamberg et al. [64]. EC was measured by employing a 4 items scale developed by Cruz and Manata [65]. EAT was measured using a 5 items scale developed by Fox & Xu [62]. PEB was measured by adapting 5 items from Juvan and Dolnicar [66] and Straughan and Roberts [67,68].

*4.3. Control Variables*

To remain consistent with existing research, this study has used demographic variables (age, gender, education) as control variables. These variables have also been used in past researches as demographic variables and have been found to influence tourist green behavior [68]. This also facilitates the researcher to avoid statistical confounds [69].

*4.4. Analysis*

SEM was employed to examine and test the theoretical model proposed in this study through AMOS v24 software [70]. This technique is more graphical user interface friendly and has been employed in prior marketing studies [71,72]. It is a two-step procedure that involves evaluation measurement and structural models [73].

**5. Results**

*5.1. Reliability, Validity, and Measurement Model Tests*

We checked the reliability and validity of measurement variables with SPSS 18.0 before testing the hypotheses. Factor loadings of all the items is above 0.70. Cronbach's alpha was measured to ensure the internal consistency of the variables and values were within the defined threshold (i.e., $\alpha > 0.70$) [74]. The values of average variance extracted (AVE) are also above 0.50 and the corresponding composite reliability (CR) are also higher than the AVE [70]. Table 2 shows the factor loadings, alpha coefficient values, and convergent validity of the constructs.

Next, for the validity verification, a confirmatory factor analysis of the measurement model was performed using AMOS v24. The fit of the measurement model was deemed acceptable with CMIN/DF = 1.897, SRMR = 0.052, GFI = 0.959, CFI = 0.975, TLI = 0.975, and RMSEA = 0.046. All the values are in the acceptable thresholds [75].

**Table 2.** Confirmatory factor analysis and reliability and validity.

| Variables | Items | Factor Loadings | Chronbach Alpha $\alpha$ | CR | AVE |
|---|---|---|---|---|---|
| Environmental Concern | EC1 | 0.795 | 0.828 | 0.834 | 0.578 |
| | EC2 | 0.714 | | | |
| | EC3 | 0.875 | | | |
| | EC4 | 0.839 | | | |
| Environmental Awareness | EA1 | 0.703 | 0.86 | 0.867 | 0.692 |
| | EA2 | 0.876 | | | |
| | EA3 | 0.895 | | | |
| Environmental Attachment | EAT1 | 0.733 | 0.834 | 0.889 | 0.617 |
| | EAT2 | 0.737 | | | |
| | EAT3 | 0.794 | | | |
| | EAT4 | 0.705 | | | |
| | EAT5 | 0.714 | | | |
| Pro-Environmental Behavior | PEB1 | 0.767 | 0.808 | 0.851 | 0.534 |
| | PEB2 | 0.815 | | | |
| | PEB3 | 0.808 | | | |
| | PEB4 | 0.812 | | | |
| | PEB5 | 0.817 | | | |

The VIF values ranging between 1.310 and 1.783 and the tolerance values between 0.835 and 0.867 indicated absence of a multicollinearity issue. Furthermore, Table 3 indicates the discriminant validity.

**Table 3.** Discriminant validity.

| | MSV | MaxR (H) | EAT | PEB | EC | EA |
|---|---|---|---|---|---|---|
| **EAT** | 0.322 | 0.894 | 0.786 | | | |
| **PEB** | 0.322 | 0.857 | 0.567 *** | 0.731 | | |
| **EC** | 0.136 | 0.943 | 0.365 *** | 0.328 *** | 0.76 | |
| **EA** | 0.161 | 0.929 | 0.380 *** | 0.402 *** | 0.369 *** | 0.832 |

Note: Significance of Correlations: *** $p < 0.001$.

*5.2. Structural Equation Modeling Analysis*

First, a structural equation modeling analysis was conducted using the maximum likelihood method. We obtained the following measurement model fitness values: CMIN/DF = 2.235, RMR = 0.052, GFI = 0.959, CFI = 0.981, TLI = 0.975, and RMSEA = 0.054. As such, the fit criteria presented by Hair et al. [72] were satisfied overall. The above results indicate that the data adequately explain the theoretical correlation between the variables used in this study. A path analysis model was then used to test the hypotheses; all the hypotheses were accepted and the results of which are shown in Table 4.

**Table 4.** Hypothesis results.

| | Estimates | S.E | C.R | *p*-Value |
|---|---|---|---|---|
| **Hypothesis 1** | 0.275 | 0.042 | 6.604 | *** |
| **Hypothesis 2** | 0.295 | 0.043 | 6.842 | *** |
| **Hypothesis 3** | 0.332 | 0.053 | 6.213 | *** |
| **Hypothesis 4** | 0.417 | 0.041 | 10.107 | *** |
| **Hypothesis 5** | 0.234 | 0.041 | 5.674 | *** |

Note: *** $p < 0.001$.

The values of the mediation estimate shown in Table 5 that relationships have a positive effect on each other. In our data, IV is environmental awareness, the mediator is environmental concern, and the DV is pro-environmental behavior, also shown in Figure 2.

**Table 5.** Direct and indirect effects.

| Hypothesis | Direct Effect | Indirect Effect | Results |
|---|---|---|---|
| EA→EC→PEB | 0.355 *** | 0.89 *** | Partial Mediation |

*** $p \leq 0.001$.

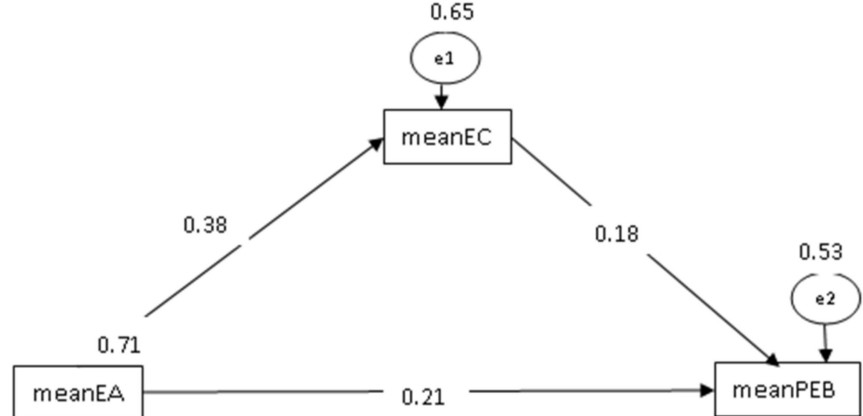

**Figure 2.** Mediation Analysis.

Hierarchical linear regression was used to test the moderating effects where we had checked three different models. In model 1 we tested the relationship between EA and PEB (β = 0.417, S.E 0.041 and $p < 0.001$) and found a significant positive relationship. Next, we checked the relationship between EA and PEB (β = 0.234, S.E 0.043 and $p < 0.001$); also, a significant positive relationship. Lastly, we checked the interaction between EA and EAT with PEB (β = 0.165, S.E 0.04 and $p < 0.001$) and results are showing significant positive relationship. Then we took the standardized values to check the strength of the relationship as shown in Figure 3. The interaction effects of environmental attachment (EAT) on environmental awareness (EA) and pro-environmental behavior. Results clearly indicates that when tourists have less environmental awareness, their pro-environmental behavior is also low. Once they get attached to the environment, then this relationship is strengthened, and this has an increased effect on the pro-environmental behavior of the tourists.

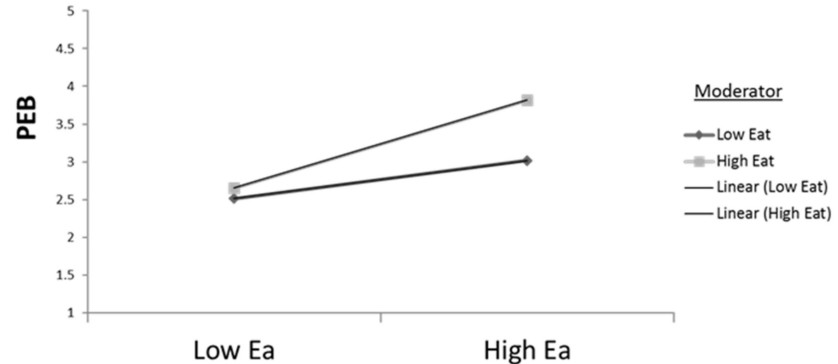

**Figure 3.** Interaction effects of environmental attachment (EAT) on environmental awareness (EA) and pro-environmental behavior.

## 6. Discussion

### 6.1. Discussion

There is a dire need to review the relationship between human and nature because of the threats posed by the environmental challenges [76]. Studying tourists' pro-environmental behaviors and attitudes is significant because they can guide scholars, managers, and policymakers related to environmental issues. Accordingly, present research investigated the antecedents of PEBs about the Pakistani tourism sector. The results of the present research held the proposed framework by suggesting key findings.

First, findings from this study suggested the significance of EA in motivating tourists to exhibit PEBs. The studies in the recent past [77] have also confirmed that customers with high EA are more inclined towards exhibiting PEBs. Second, findings from this study also confirm the significance of EA in predicting EC. This finding suggests that the tourists with high EA are more concerned regarding environmental issues and challenges. Moreover, when individuals have appropriate EA, they become more concerned with regard to environment protection and sustainability [78]. The studies in the past [44] have also reported the similar results. Third, the results of the current research confirm that tourists with high EC are more inclined towards exhibiting PEBs. Deep concerns for environmental issues have led individuals to exhibit environmentally responsible behaviors [79]. Notable scholars [80,81] have also reported the similar findings. Fourth, this study empirical finding also confirms the mediation of tourists' EC in linking EA with PEBs. The interaction of individuals' sense of responsibility and concern regarding outcomes of their activities (EA) with their sense of community membership can activate their moral obligation, resulting in PEBs [82]. Finally, this study found that environmental attachment moderated the relationship of environmental awareness and pro-environmental behavior. The results clearly indicated that when tourists are less attached to the environment, even by increasing their awareness to the environment, it will not have any significant change in their pro-environmental behavior. On the contrary, for the tourists who are attached to the environment, a slight increase in their environmental awareness will increase their pro-environmental behavior exponentially.

### 6.2. Managerial Implications

The present research also offers valuable managerial implications. Environmental management has become significant in developing nations such as Pakistan because majority of tourism destinations face rising environmental challenges and issues [83]. The natural resources of Pakistan have been under terrific pressure because of unrestricted tourism activities, coupled with unsustainable exploitation of natural resources by tour operators and tourists. This research suggests that challenges and issues related to the environment can be overcome or mitigated if the tour operators and firms related to tourism promote travelers' green behaviors by enhancing their EA. Environmental awareness and concern towards its protection can facilitate tourists in conserving resources by exhibiting PEBs; hence, protecting and positively influencing the natural environment. The present research demonstrates that a significant change in tourists' behavior with regard to greening of destination visited can only be brought when the right attitude towards the environment (EC) is being possessed by them. Moreover, the absence of EA can impede green initiatives; hence, it is suggested that tour operators should educate tourists on various environmental issues and challenges being faced by popular travel destinations because of human actions and activities, thus enhancing their EA. The tourists are highly expected to exhibit PEBs when they are both aware and have concern regarding the impact of human activities on environment.

Tourists are visiting northern areas due to the environment of the tourist spots, and the government agencies working on tourism should promote neat and clean images so a positive image for tourism in Pakistan can be portrayed. Furthermore, keeping the environment neat and clean is also helpful for promoting green environment in Pakistan. Pakistan is also becoming a well-known tourist spot for foreign tourists. Foreign tourists

are visiting the northern areas, and Gwadar is also a hotspot for tourists currently. The government should take necessary measures to preserve the natural beauty of the tourist spots to maintain a good for tourism in Pakistan.

### 6.3. Limitations and Future Research Directions

This study has limitations, which should be addressed in further research. First, the small sample size is a limitation. Future research may benefit from a larger sample size in order to produce more valid findings. Second, the research is carried out on Kumrat Valley experiences only. Thus, research should be conducted on other northern areas and other tourist spots of Pakistan as well. In this study, we only present the awareness–behavior gap, without empirically investigating the differences in impact between barriers and motivators on pro-environmental behavior, leaving significant scope for further exploration. All respondents to our questionnaire are tourist who have visited Kumrat Valley. Future studies could identify differences between other tourism spots environmental awareness and pro-environmental behavior by obtaining more samples from more diverse populations. Finally, environmental awareness may have different effects on different pro-environmental behaviors. Further studies can be on the other tourist spots in Pakistan. Further research can specifically examine the effects of awareness on different behaviors in tourism, such as green behavior and religiosity. Moderator can be used within other boundaries, for instance, between EA and EC instead of EA and PEB, the direct path.

### 7. Conclusions

Kumrat Valley has become a tourist hotspot in recent times. The major environmental issues include air/water pollution, natural resource depletion, climate change, waste disposal, biodiversity loss, and overcrowding. Littering in the valley can cause pollution and changes in climate. As an important economic sector, the travel and tourism industry can play an important role in handling environmental change. The level of environmental awareness outweighed pro-environmental behavior. The mediation and moderation in this study are strengthening the relationship between awareness and behavior. The study has limitations, which should be addressed in further research. The small sample size is a limitation. Future research may benefit from a larger sample size in order to produce more valid findings. The research is carried out on Kumrat Valley experiences only. Thus, research should be conducted on other northern areas and other tourist spots of Pakistan as well. In this study, we only present the awareness–behavior gap without investigating the differences in impact between barriers and motivators. Future studies could identify differences between other tourism spots, environmental awareness, and pro-environmental behavior. Further research can specifically examine the effects of awareness on different behaviors in tourism.

**Author Contributions:** Conceptualization, S.A., Z.A. and N.M.H.; methodology, Z.A., N.M.H., M.N.K., M.A.M. and M.F.; software, Z.A.; validation, M.A.M. and M.F.; formal analysis, S.A., Z.A. and N.M.H.; investigation, M.F.; resources, M.N.K., M.A.M. and M.F.; data curation, N.M.H. and M.A.M.; writing—original draft preparation, S.A., Z.A. and N.M.H.; writing—review and editing, M.N.K., M.A.M. and M.F.; visualization, M.A.M. and M.F.; supervision, Z.A.; project administration, Z.A., N.M.H. and M.N.K.; funding acquisition, M.N.K. All authors have read and agreed to the published version of the manuscript.

**Funding:** This research received no external funding.

**Institutional Review Board Statement:** Not applicable.

**Informed Consent Statement:** Informed consent was obtained from all subjects involved in the study.

**Data Availability Statement:** The data presented in this study are available on request from the corresponding author.

**Acknowledgments:** This research was supported by the Ajman University, Ajman, UAE.

**Conflicts of Interest:** The authors declare no conflict of interest. The funders had no role in the design of the study; in the collection, analyses, or interpretation of data; in the writing of the manuscript; or in the decision to publish the results.

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
