# Peer review of "Impact of Tourist’s Environmental Awareness on Pro-Environmental Behavior with the Mediating Effect of Tourist’s Environmental Concern and Moderating Effect of Tourist’s Environmental Attachment"

_sustainability, doi:10.3390/su132312998_

Round 1

Reviewer 1 Report

The study examines the pro-ecological behavior of travelers, including those promoting environmental protection
and ecosystems harmful to avoidance, as well as choosing modes or products that are
environmentally friendly.
The structure of the scientific article was preserved, the results were supported by a graphic presentation in the form of tables and charts.
An interesting discussion ends the article. There is an urgent need to revise the relationship between man and nature, due to the threats posed by environmental challenges. Research on tourists' pro-environmental behavior and attitudes is important as they can guide scientists, managers and policy makers related to environmental issues. The results of this study confirmed the proposed framework, suggesting key conclusions. Valuable suggestions have been made for further research in this area.

I suggest making some minor adjustments to make the article even better:
- change your mind: "Environmental awareness requires not only environmental awareness, but also the mindset, beliefs, and skills of a person to solve environmental problems..."
- improve the uniformity and readability of tables - it is worth paying attention to the same format of tables and explanations on graphic elements

- correct punctuation in the bibliography.

Author Response

First, we are grateful for your time and efforts to improve our paper.

Review comment- change your mind: "Environmental awareness requires not only environmental awareness, but also the mindset, beliefs, and skills of a person to solve environmental problems..."

Author’s Response -We agree with the worthy reviewer’s suggestion. We have broadened the scope of “Environmental Awareness” and added two more definitions to dig deep into the concept of environmental awareness in pursuit of incorporating the aspects which above mentioned & have guided us to inculcate.

Author’s Comments: The studied literature review divides it into two streams, first talks about a set of affections and understanding related to human behaviour and environment [26], whereas the second describes it as an ability to integrate sensations from the environment and related issues with targeted goals to generate pro-environmental behavior [20].

Done “Scope of Environmental awareness” has been modified. Page 6-line no. 12-15.

Review comment- improve the uniformity and readability of tables - it is worth paying attention to the same format of tables and explanations on graphic elements

Author’s Response - Tables have been reformatted uniformly whereas figures have also been adjusted accordingly.

Author’s Comments: Done

- correct punctuation in the bibliography.

- bibliography has been adjusted according to journal requirements.

Author’s Comments: Done

Reviewer 2 Report

Dear Authors. This paper is well written.

Author Response

Thank you very much for appreciating our efforts.

Reviewer 3 Report

Brief summary

The study examines the mediating role of environmental concern between 
environmental awareness and pro-environmental behavior, as well as the 
moderating effect of environmental attachment.

Specifically, the Authors illustrate and test five hypotheses: 

  • Environmental awareness positively influences tourists pro environmental behaviour;
  • Environmental awareness positively influences environmental concerns;
  • Environmental concern positively influences tourists pro environmental behaviour;
  • The relationship between environmental awareness and pro environmental behaviour is mediated by environmental concern;
  • The relationship between environmental awareness and pro environmental behaviour is moderated by environmental attachment.

The empirical analysis is based on data collected through 426 questionnaires. Five-point likert scale has been employed to assess the study variables (EA, EC and PEBs). EA is measured by employing 3 items,
EC is measured by employing 4 items, PEB is measured by adapting five items. Demographic variables are used as control variables. SEM is employed to examine and test the theoretical model proposed in this study. According to the Authors, data explain the theoretical correlation between variables, then through a path analysis model all hypotheses are tested and not rejected. Results indicate that when tourists have less environmental awareness their pro-environmental behavior is also low. Once they get attached to the environment, they also improve their awareness and consequently their pro-environmental behavior.

Lying on the results of the analysis, the Authors suggest that challenges and issues related to environment can be mitigated by incentivizing tour operators to promote travelers green behaviors by enhancing their environmental awareness, as the latter can motivate tourists in preserving resources by exhibiting PEBs hence protecting and positively influencing the natural environment. A significant change in tourists’ behavior with regard to greening of destination visited can only be brought when a right attitude towards environment (EC) is developed. 

Broad comments

The manuscript illustrates an interesting empirical research on a complex relationship involving two constructs (Environmental awareness and pro-environmental behaviour), one mediator (environmental concern) and one moderator (environmental attachment). However, there are numerous aspects of the research that should be improved.

First, in the introduction, the numerous topics illustrated should be more interconnected, and the research purpose should be clearly stated. Specifically, the discourse begins by introducing the relevance of tourism within the world economy, then shifts on tourism attractiveness of Pakistan and Kumrat. Then, the role of the travel & tourism industry is analyzed, and the relevance of the pro-environmental behaviour is introduced. Finally, tourists' behaviour is connected to environmental change, and the variables analyzed more in depth in the empirical research are introduced.

Given the complexity of the narrative, at least few sentences (or a graph) may be added to summarize the many passages. Then, the research objective may be introduced. Alternatively, Authors are invited to summarize the introduction in one page and pospone a more detailed discussion in the section dedicated to theoretical background, that, instead, lacks of more in depth analyses.

Concerning the Theoretical background, the paragraph dedicated to the analysis of the ABC model seems disconnected from the rest of the section, where the variables included in the empirical anlysis are illustrated more in detail. Few sentences explaining why the ABC model is introduced may help to bridge the gap.    

In section "Materials and Methods", paragraph 4.2. "Measures" illustrates the number of items used to measure each variable. A table illustrating the label associated to each item  and its connection with the aggregate constructs may allow the reader to better understand how each variable has been obtained. 

In paragraph 5.1. "Reliability, validity and measurement model test" VIF and tolerance are not reported in table 2 (VIF is only mentioned within the text).

In paragraph 5.2. the output of the SEM analysis should be illustrated more in detail and a graph similar to Figure 1 reporting relations and coefficients may facilitate the reader in interpreting results, as well as a computation of the total, direct and indirect effects. Figure 2 needs further explanation, as Authors do not mention the procedure that leads to the results illustrated. 

Specific comments

The following setence in section 5.2. is not clear: "The following results were obtained for Korean and multinational pharmaceutical companies". Authors are invited to clarify its meaning.

Author Response

First of all, we are really grateful for your time and efforts to critically review our paper in an attempt to improve our paper quality.

Review comment 1- at least few sentences (or a graph) may be added to summarize the many passages. 

Author’s Response – Current study is a significant addition to the swelling body of knowledge on environmental concerns and pro-environmental behaviors exhibited by the tourists, both domestic and international. Therefore, the objective of this paper is to explore the unique and novel construct from the perspective of a developing country and to explore the intervening mechanism of a tourist’s environmental attachment. Secondly, the research integrates the ABC model by Albert Ellis (1995) to examine the disparity in the attitude-behavior of tourists visiting the North of Pakistan specifically Kumrat valley. Concluding paragraph to summarize the introduction paragraph has been added on Page 5-line no. 30-33 & Page 6-line no. 1-3.

Review comment 2- Few sentences explaining why the ABC model is introduced may help to bridge the gap.    

Author’s Response – The beauty of this model lies in predicting how individuals feel or will react towards a particular object/thing or event. In our study Environmental awareness is the activating event (A), as the awareness of an individual can help in reducing the pollution in his nearby environment. After creating awareness in the mind of an individual, the sense of environmental concern is developed then from which an individual forms a belief (B), to protect the environment and becomes environmentally concerned which results in the consequences (C), on the pro-environmental behaviors due to this belief. The consequences can be positive or negative according to pro-environmental behavior. Reasons to select ABC model & its relevance to the given context has been incorporated on Page 8-line no. 7-14.

Review comment 3- In the section "Materials and Methods", paragraph 4.2. "Measures" illustrates the number of items used to measure each variable. A table illustrating the label associated with each item and its connection with the aggregate constructs may allow the reader to better understand how each variable has been obtained. 

Author’s Response we have created a new Appendix file to show the items of each variable, and how they form the aggregate construct along with their references. We are hopeful that this will address the query.

Review comment 3 - In paragraph 5.1. "Reliability, validity, and measurement model test" VIF and tolerance are not reported in table 2 (VIF is only mentioned within the text).

Author’s Response Values of VIF and Tolerance have been mentioned separately just to give an idea that there is no multicollinearity problem in the data. Furthermore, Table 3 only presents the discriminant validity. We are hopeful that this data presentation is sufficient for the readers.

Review comment 4 - In paragraph 5.2. the output of the SEM analysis should be illustrated more in detail and a graph similar to Figure 1 reporting relations and coefficients may facilitate the reader in interpreting results, as well as a computation of the total, direct and indirect effects. Figure 2 needs further explanation, as the Authors do not mention the procedure that leads to the results illustrated.

Author’s Response A figure related to mediation analysis has been added on Page 15-line no. 1 along with a description on Page 14-line no. 16,17 & 18.

.

Review comment 5 - The following sentence in section 5.2. is not clear: "The following results were obtained for Korean and multinational pharmaceutical companies". Authors are invited to clarify its meaning.

Author’s Response We are really sorry for mistakenly adding the above-mentioned sentence in this manuscript. Currently, we are working on another research paper related to “Physician’s prescription behavior” this line belongs to that project and had been mistakenly posted here. It has nothing to do with this research paper. Kindly ignore it and accept the amended version of the document.

Round 2

Reviewer 3 Report

Broad comments

The revised version of the manuscript satisfies most of the issues mentioned in the first round of revision. Although the quality of the presentation can be improved, each element has now been clearly defined in the supplementary material and the main output of the quantitative analysis has been illustrated in Figure 2 (however, the correlations shown in Figure 2 cannot be found in the paragraph). Although a clear description of how figure 3 was obtained is not given, this is a standard output, so it can be assumed that a standard process was used.

Authors are kindly invited to illustrate in a table at least the regression output that led to the impacts indicated in Figure 2, and to make an additional effort in illustrating the process that led to the representation illustrated in Figure 3.

Author Response

  1. A table (as shown below) illustrating the regression output has been inserted.

Table 5. Direct and Indirect effects

Hypothesis

Direct effect

Indirect effect

Results

EAàECàPEB

0.355***

0.89***

Partial Mediation

***p=<0.001

  1. To illustrate Figure 3 a detailed discussion (as given below) of the process has been added in the draft.

“Hierarchical linear regression was used to test the moderating effects where we had checked three different models. In model 1 we tested the relationship between EA and PEB (β=0.417, S.E 0.041 & p<0.001) and found a significant positive relationship. Next, we checked the relationship between EA and PEB (β=0.234, S.E 0.043 & p<0.001) also a significant positive relationship. Lastly, we checked the interaction between EA and EAT with PEB, and (β=0.165, S.E 0.04 & p<0.001) was also showing a significant positive relationship. Then we took the standardized values to check the strength of the relationship as shown in Figure 3.”

This manuscript is a resubmission of an earlier submission. The following is a list of the peer review reports and author responses from that submission.

Round 1

Reviewer 1 Report

  1. It would be worthwhile to discuss the methods used and the results obtained by these methods in a little more detail in Sections 5.1 and 5.2. what follows from the recommendations to the authors . Presently, they are discussed too laconically.
  2. 4.2 . Measurement of the variables  - whether the questions included in the questionnaire addressed to tourists concerned pro-ecological behaviour mainly in their everyday life or also during tourism - it would be useful to describe the questionnaire a bit more broadly 
  3. The passage describing the spatial extent and characteristics of the study area (introduction - lines 58 to 99) would fit better in section 4
  4.  Is there a difference/ similarity between the results of the conducted survey (the degree of tourists' environmental awareness and their willingness to act pro-environmentally) and the actual state of the environment observed in the studied area. The question can be asked as to why the survey was conducted in this particular area - are we dealing with more aware, responsible tourists here? 

Reviewer 2 Report

Dear authors, I found you work very interesting. Hovewer I propose you some suggestion to improve it.

Abstract

Try to reduce the length focusing on the main issue, the problem statement, the main results and benefit of your research. A short and good written abstract attract the reader.

Introduction

You started introducing the tourism sector but I don't found any reference to this sector in title and abstract. I suggest to improve putting the focus on this industry.

Add the purpose statement in the introduction section clarify the research premise that drive your study. At the end explain the structure of the manuscript.

Theoretical background

improve this section explain the reason that lead you to consider the elements in sub section2.1 - 2.5. Why these and not others?  Improve the theoretical background fostering the references.

Hypotesis

Add the discussion of figure 1

Materials and methods

Justifies with references lines 331-350. Is your sample representative referring to the population? Improve the questionnaire description and provide the questions as item in a table or as appendix.

Results 

discusse the results coming from Crombach's Alpha.

Discussion

Improve the discussion section linking the results to the issues in tourism sector. Managerial implications are weak. Add theoretical implications. why is your study important? what does it add to the scientific landscape?

Generally I suggest you to improve English language using a professional service and review the formatting. 

Reviewer 3 Report

The article claims to uncover gaps in environmental awareness–behavior considering environmental awareness and pro-environmental behavior. The article is crudely written and lacks coherence. The texts are also written in a very confusing manner. It was hard to follow the arguments made by the authors. The exact research question the authors are trying to address by performing this study is not clear from the text. The distribution of the text is very uneven. The authors tried introducing the background and theoretical context in a large proportion of this paper with a lack of evidence on various occurrences. Even the abstract written is not clearly communicating the message of their work. The article also contains many grammar errors, making it much harder to follow the semantic flow of text.  

Even following through, the introduction of the touristic places (Line 51-104) seems more like a commentary and less relevant to the study design. The authors missed significant literature to back the claims the paper made in those lines. The authors skipped the part of explaining and connecting the relation to what they have discussed in the initial part of the paper in the method and result section. It is not clear to me how to follow and understand the aim and outcome of this paper without concrete research question(s).